# Ecological Quality Assessment of Greek Lowland Rivers with Aquatic Macrophytes in Compliance with the EU Water Framework Directive

Konstantinos Stefanidis [1,2,*], Georgios Dimitrellos [1], Maria Sarika [3], Dionysios Tsoukalas [1] and Eva Papastergiadou [1,*]

[1] Department of Biology, University of Patras, University Campus, 26504 Rio, Greece
[2] Hellenic Centre for Marine Research, Institute of Marine Biological Resources and Inland Waters, 46.7 km of Athens—Sounio Ave., Anavyssos, 19013 Attiki, Greece
[3] Department of Biology, National and Kapodistrian University of Athens, 15701 Athens, Greece
[*] Correspondence: kstefani@upatras.gr (K.S.); evapap@upatras.gr (E.P.); Tel.: +30-229-107-6439 (K.S.)

**Abstract:** Aquatic macrophytes are one of the four biological quality elements (BQE) used for assessing the ecological status of inland waters according to the EU Water Framework Directive (WFD 2000/60). With this article, we present the methodological approach for the implementation of a WFD compliant macrophyte index to the riverine systems of Greece. In addition to the definition and harmonization of the ecological quality class boundaries, the results from the pilot application of the index and the ecological classification of the monitored river reaches are also presented. Aquatic plants and environmental parameters were sampled from 93 river reaches between 2012 and 2015. A multivariate analysis with optimal scaling (MVAOS) was conducted to define the main stressor gradient and to identify the least disturbed sites and the reference conditions that are required for the derivation of the ecological quality classes. The Macrophyte Biological Index IBMR for Greek rivers (IBMR$_{GR}$) was calculated for all the sites and the boundaries for the five quality classes were derived according to the methodology proposed by the Mediterranean Geographic Intercalibration Group (MedGIG). The main findings showed that the hydromorphological modifications were the main environmental stressors that correlated strongly with the IBMR$_{GR}$, whereas physicochemical stressors were of lesser importance. More specifically, the first principal component explained 51% of the total variance of the data, representing a moderately strong gradient of hydromorphological stress, whereas the second component explained 22.5%, representing a weaker gradient of physicochemical stress. In addition, the ecological assessment showed that almost 60% of the sites failed the WFD target of the "Good" ecological quality class, which agrees with classification assessments based on other BQEs for Greece and many Mediterranean countries. Overall, this work provides a first assessment of the ecological classification of Greek rivers with the BQE of aquatic macrophytes with significant implications for ecological monitoring and decision making within the frame of the WFD implementation.

**Keywords:** aquatic macrophytes; Water Framework Directive; rivers; ecological quality; ecological monitoring; Eastern Mediterranean





## 1. Introduction

Aquatic macrophytes are aquatic photosynthetic organisms easily seen with the naked eye and include vascular plants, mosses, liverworts and macro-algal growths [1]. They have been widely used as bioindicators in freshwater habitats because certain species and communities are known to respond to environmental changes caused by anthropogenic perturbations such as eutrophication, acidification and hydromorphological alteration [2–6]. Several studies have investigated the role of anthropogenic disturbances in shaping the structure and functioning of macrophyte communities [7–10] revealing various and complex responses of diversity and community indices to gradients of hydromorphological

features and nutrients. It is well known that numerous/multiple human activities such as agriculture, aquaculture, urban infrastructure and settlements, alterations in the hydromorphology and flow regime, significantly influence the abundance, structure and the extent of the macrophyte communities [11,12]. Naturally, aquatic macrophytes were recognized as an important tool for biomonitoring and assessment of freshwater ecosystems and were adopted as one of the four biological quality elements (BQEs) that are used for the ecological classification of streams and rivers in Europe, following the implementation of the Water Framework Directive (WFD 2000/60) [13].

The goal of the Water Framework Directive [14] is to restore or maintain good ecological state of freshwater systems of all EU member states. Thus, the WFD provides very detailed guidelines for the implementation of the ecological monitoring and the assessment of all European inland and coastal waters, including rivers and streams. The ecological monitoring and assessment involve the monitoring of biological, hydromorphological and physicochemical quality elements. For running waters (rivers and streams) the goal of the "Good" ecological status is defined in Annex V of the WFD and refers to terms of quality assessed with the use of biological communities, based mainly on diatoms, benthic invertebrates, fish and aquatic macrophytes. Basically, the ecological status is derived by comparing the biological community of a certain site with the respective community that would be expected in environmental conditions with no or minimal anthropogenic impact. These conditions are known as reference conditions and can be defined using different approaches [15].

Today, numerous biological assessment systems based on macrophytes have been developed and used by EU members [1]. Most of these systems are based on indices that consider the species composition of macrophyte assemblages and species indicator values that reflect the tolerance to a certain disturbance (e.g., organic pollution) [16,17]. The Macrophyte Biological Index for Rivers (IBMR) is one of these indices originally developed for France [17] and adopted by other EU members (e.g., Portugal, Italy, Cyprus and Greece) [1,13]. The calculation of the IBMR is based on indicator taxa that belong to various macrophyte groups, such as macroalgae (e.g., Characeae), aquatic bryophytes (e.g., *Fontinalis* sp.), truly aquatic macrophytes (e.g., *Potamogeton* sp.) and emergent vascular species (e.g., *Polygonum* sp.) [1,17]. In Greece, the IBMR$_{GR}$ is the national assessment method for classification of ecological quality of rivers with the use of macrophytes that has been intercalibrated during the Mediterranean Geographic Intercalibration Group exercise (MedGIG) [1] and it has been implemented during the first round of the National monitoring program (2012–2015) and the second phase which started on 2018 and is still running [18]. The IBMR$_{GR}$ list of indicator species included new species that are characteristic of the Greek rivers to adjust the index in the local conditions. Earlier studies on the aquatic macrophyte communities of the Greek riverine ecosystems have focused on specific rivers examining mostly associations between plant communities and environmental gradients [19,20]. A more recent study by Stefanidis et al. [2] examined the biodiversity patterns of aquatic macrophytes across environmental gradients at a larger spatial scale covering multiple catchments and sampling sites, which are included in the current study.

Driven by the lack of a national ecological assessment method for Greek rivers with the use of aquatic macrophytes, the current study is the first ever that focuses on the development and use of a macrophyte index as an official WFD-compliant national method. More specifically, this paper describes the methodological approach for the implementation of the IBMR index to the riverine systems of Greece, including the multi-step procedure for the definition and harmonization of the ecological quality class boundaries. It also provides a first ever detailed overview of the ecological classification of the riverine ecosystems of Greece based on macrophytes that resulted from the implementation of this pilot monitoring scheme during the first phase of the ecological monitoring (2012–2015) within the frame of the WFD 2000/60.

## 2. Materials and Methods

### 2.1. Field Surveys

Field surveys were conducted in a total of 93 stream sites (Figure 1), which are part of the National Monitoring Network [21], covering all the biogeographic regions of mainland Greece [22]. Samplings were conducted once between April and August of 2014 and 2015 following standardized protocols from the MedGIG [1]. Macrophytes were sampled from both banks and the channel when feasible, by wading upstream for a 100 m long section of the river reach. The abundance of each species was assessed using a 5-point cover scale where 1 stands for rare plants with cover ranging from 1 to 5%, 2 for occasional plants with cover 6–25%, 3 for frequent plants and cover 26–50%, 4 for abundant plants with cover between 51 and 75% and 5 for dominant plants with cover between 76 and 100%. Most species were identified in the field, but some specimens were collected and transferred to the laboratory for further identification. A complete list with the identified plant taxa can be found in the supplementary material (Table S1).

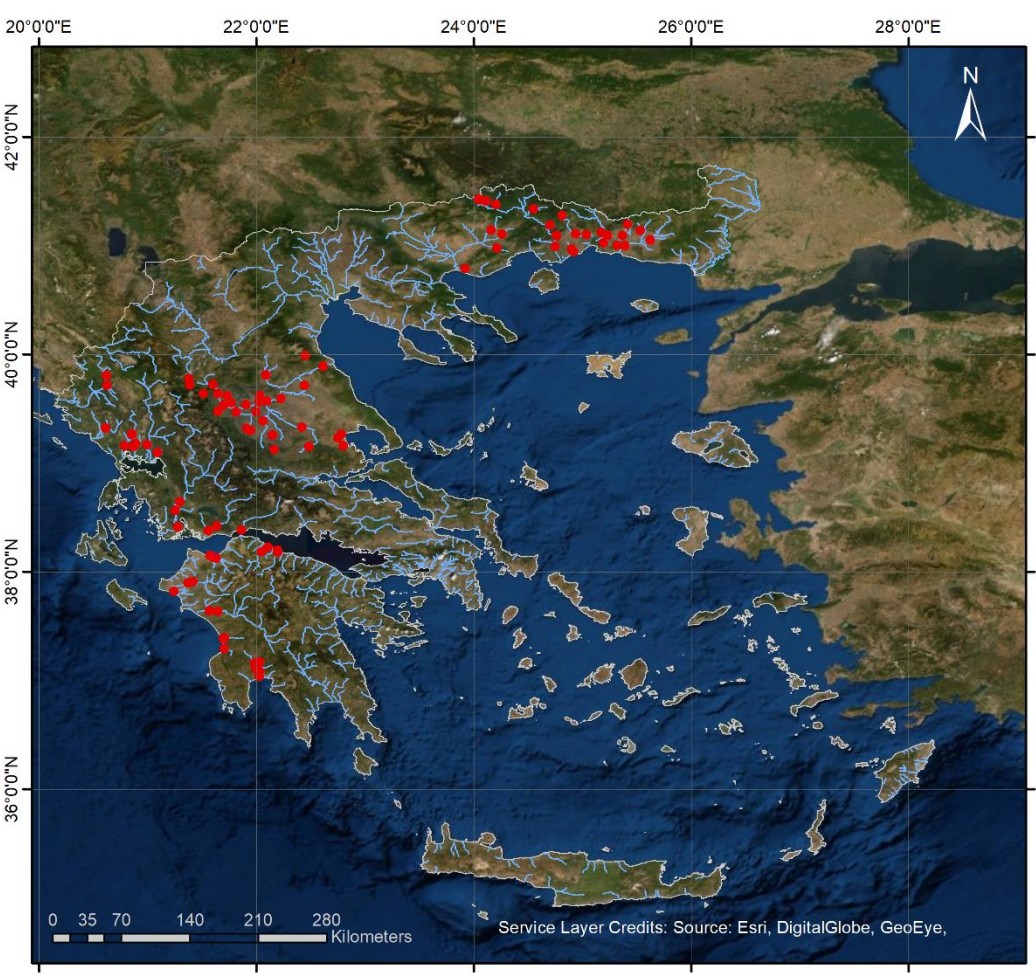

**Figure 1.** Location of sampling sites (*n* = 93) of the National Monitoring network, across streams and rivers of mainland Greece.

Water was sampled and transferred to the laboratory for the chemical quantification of orthophosphates, nitrogen species (nitrate, nitrite, and ammonium concentrations in water), total inorganic nitrogen and total phosphorus following the analytical procedures according to APHA [23]. Electrical conductivity, water temperature, dissolved oxygen, and pH were measured in site with a portable multi-meter probe. In parallel with the macrophyte and water sampling, hydromorphological characteristics such as channel cross section alteration, water abstraction, presence of dykes, hydrological alteration,

etc., were recorded as described by Feio et al. [15] (see Table 1). Both physicochemical and hydromorphological variables were used as proxies of anthropogenic stressors (e.g., nutrient pollution, acidification, hydromorphological modifications) in order to define the main stressors that influence the macrophyte index.

**Table 1.** Environmental variables that were considered as potential stressor indicators.

| | **Stressor** | **Description** |
|---|---|---|
| **Hydromorphological** | Channel profile/cross section alteration | Degree of channel profile modification present at the site/cross section alteration |
| | Channel morphology | Degree of the morphological modification of the channel present at the site |
| | Local habitat alteration | Alteration of instream habitats |
| | Stream hydrology | Degree of the hydrological alteration present at the site |
| | Upstream dams influence | Effect of upstream dams |
| | Water abstraction | Effect of water abstraction at the site |
| | Dykes (flood protection) | Effect of dykes for flood protection |
| **Physicochemical** | pH | Sorensen scale |
| | Conductivity | Conductivity [mS/cm] |
| | Ammonium | Ammonia concentration in the water [mg/L $NH_4^+$] |
| | Nitrate | Nitrate concentration in the water [mg/L $NO_3^-$] |
| | Total nitrogen | Total Nitrogen [mg/L TN] |
| | Total phosphorus | Concentration of total phosphorus in the water [mg/L TP] |
| | Orthophosphates | Concentration of Orthophosphates in the water [mg/L $PO_4^{3-}$] |
| | DO | Concentration of dissolved oxygen [mg/L] |
| **Land use** | Urbanization | Urban and industrial areas in immediate vicinity of site |
| | Agriculture | Agriculture at the immediate vicinity of site |

*2.2. Statistical Treatment of Environmental Pressure Data—Identification of the Main Stress Gradient*

We applied a multivariate analysis with optimal scaling (MVAOS) for the entire dataset of environmental variables (Table 1) to identify those with the highest contribution in the variance of the data. Although there are many other methods for handling multivariate data analysis in water sciences [24,25], we used MVAOS because it allows us to extend the concept of the principal component analysis (PCA) to ordinal variables by transforming them to scale variables [26]. Basically, the MVAOS procedure transforms the ordinal variables and then a PCA is conducted. After omitting those variables with a correlation coefficient r < 0.7 with the first two principal components (PC1 and PC2), a second PCA was conducted with the remainder variables. Thus, the MVAOS PCA was used for dimensionality reduction and for providing an environmental gradient as a proxy for stressor gradient. Furthermore, the PC1 scores for the sites were used to identify the less disturbed or unstressed sites which represented the reference sites according to the guidelines provided by the MedGIG. Unstressed sites were defined as those with a PC1 value less than the 25th percentile of the total scores. The MVAOS procedure was applied with the "Gifi" package [27]. PCA were then performed on the transformed variables with the "FactoMineR" package [28]. All analyses were done in R environment [29].

*2.3. Development and Implementation of the IBMR$_{GR}$ Index*

The IBMR$_{GR}$ index was calculated for all sites according to the following formula [17]:

$$\text{IBMR} = \frac{\sum_i (E_i K_i CS_i)}{\sum_i (E_i K_i)} \tag{1}$$

where $E_i$ the coefficient of ecological amplitude for a given species $i$, $K_i$ the scale of cover and $CS_i$ is the species-specific score that indicates tolerance to organic pollution.

Sites with only one or two scoring macrophyte species were excluded from further analysis leaving a total number of 79 from the initial 93 sites. In order to distinguish how well the IBMR$_{GR}$ index responds to the pressure a simple linear regression was conducted between the macrophyte index and the PC1 scores derived from the PCA on the most important stressor variables.

Then, the index is normalized between 0 and 1 as follows [30]:

$$IBRM_{NORM} = \frac{I - SI_5}{USI_{75} - SI_5} \tag{2}$$

where $I$ is the index value at a given site, $SI_5$ is the 5th percentile of the stressed sites and $USI_{75}$ is the 75th percentile of the unstressed sites. Normalized values larger than 1 are set to 1 and lower than 0 are set to 0.

To determine the boundary of the index between the High and Good ecological quality class the 25th percentile of the unstressed sites was used. For the boundaries between the other quality classes the guidelines from the Common Implementation Strategy [31] were followed and the 25th percentile value was divided to 4 so each quality class (Good, Moderate, Poor and Bad) has the same range as the others.

## 3. Results and Discussion

*3.1. Determination of the Least Disturbed/Unstressed Sites*

The results from the first MVAOS PCA including all the environmental data showed that the variables with the highest correlation with the PC1 were the channel profile/cross section alteration, channel morphology, habitat alteration, stream hydrology, water abstraction and agriculture, whereas ammonium and phosphate concentrations correlated strongly with the PC2 (Table 2). A second MVAOS PCA was conducted keeping only these nine variables and the results showed that the PC1 acts as a gradient of hydromorphological stressors whereas PC2 clearly shows a strong relationship with three physicochemical variables (ammonium, nitrate, and phosphate) (Figure 2). The first component explained 51% of the total variance of the data whereas the second component explained 22.5%. Thus, the PC1 represents a moderately strong gradient of hydromorphological stress and the PC2 a weaker gradient of physicochemical stress. Then, using the PC1 scores of the sites, that indicate their position along the first component, we distinguished the sites that are less affected by the hydromorphological stressors (those that are positioned at the left side of the biplot in Figure 2). To do so we defined as less disturbed sites (or unstressed) those with a PC1 score less than the 25th percentile of the total PC1 scores. Sites with PC1 score between the 25th and 75th percentile were considered as moderately stressed and those with a PC1 score >75th percentile were the highly stressed sites. Based on this discrimination, Figure 2 shows that unstressed sites are clearly separated as they are placed at the far left in the biplot. Moderately stressed sites are distributed along axis 1 and highly stressed sites are placed on the right part of the plot with several points also showing a high correlation with the second axis that represents a gradient of physicochemical stress. The results showed that the examined variables were responsible for a substantial portion of the variance of the dataset, indicating the importance of hydro-morphological disturbances in the macrophyte communities of the investigated stream sites. To further show the clear discrimination of the sites between these three groups of stress intensity we performed a Kruskal-Wallis test for the IBMR$_{GR}$ values among the stress level and we found that the

macrophyte index medians are significantly different ($p \leq 0.001$). Figure 3 illustrates these differences between the unstressed, moderately stressed and highly stressed sites.

**Table 2.** Correlations between the environmental variables and the first two principal components. Values > 0.7 (in bold) indicate which variables are retained for further analysis.

| Stressor | PC1 | PC2 |
|---|---|---|
| **Channel profile/cross section alteration** | **0.882** | −0.082 |
| **Channel morphology** | **0.872** | −0.099 |
| **Habitat alteration** | **0.848** | 0.012 |
| **Stream hydrology** | **0.877** | −0.093 |
| Dams influence | −0.137 | 0.35 |
| **Water abstraction** | **0.822** | −0.048 |
| Dykes | 0.619 | −0.282 |
| DO | −0.436 | −0.225 |
| pH | −0.168 | −0.21 |
| Electrical conductivity | 0.596 | −0.109 |
| **Ammonium** | 0.407 | **0.799** |
| **Nitrate** | 0.283 | **0.834** |
| **Phosphate** | 0.268 | **0.82** |
| Urbanization | 0.325 | 0.167 |
| **Agriculture** | **0.868** | −0.189 |

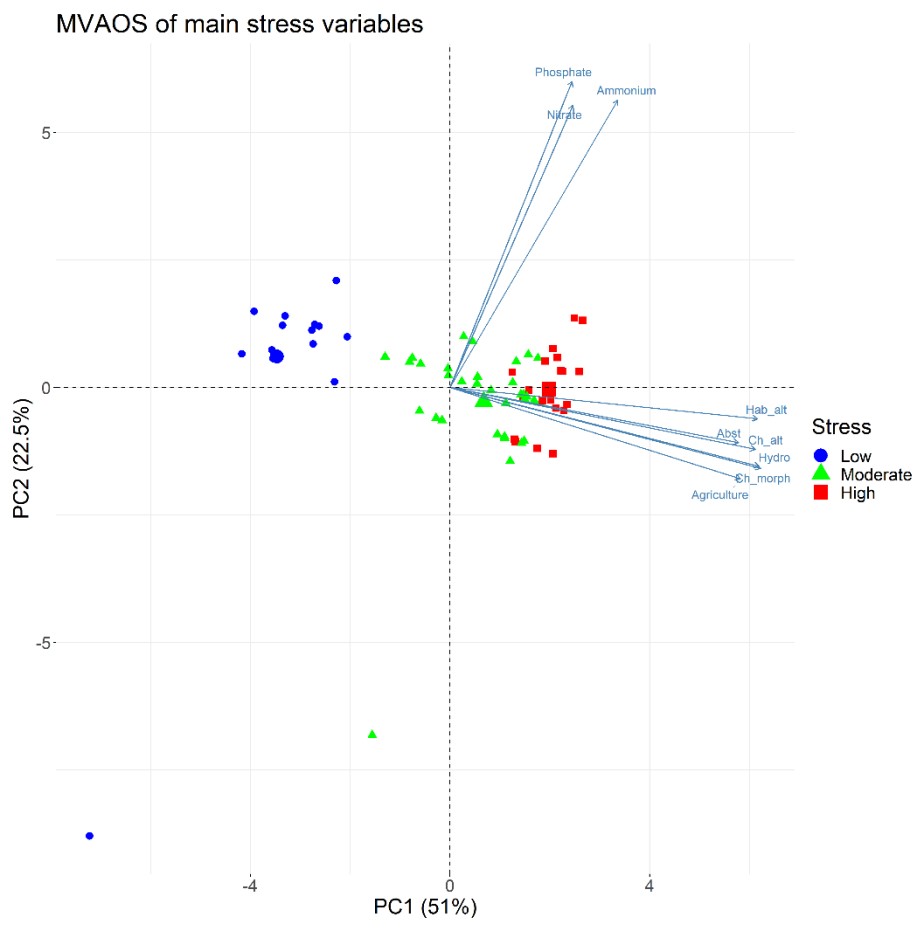

**Figure 2.** A MVAOS principal component analysis biplot with the nine most important environmental stressors. The position of the sites along the principal component 1 indicates the level or perturbation (stressed, moderately stressed and unstressed).

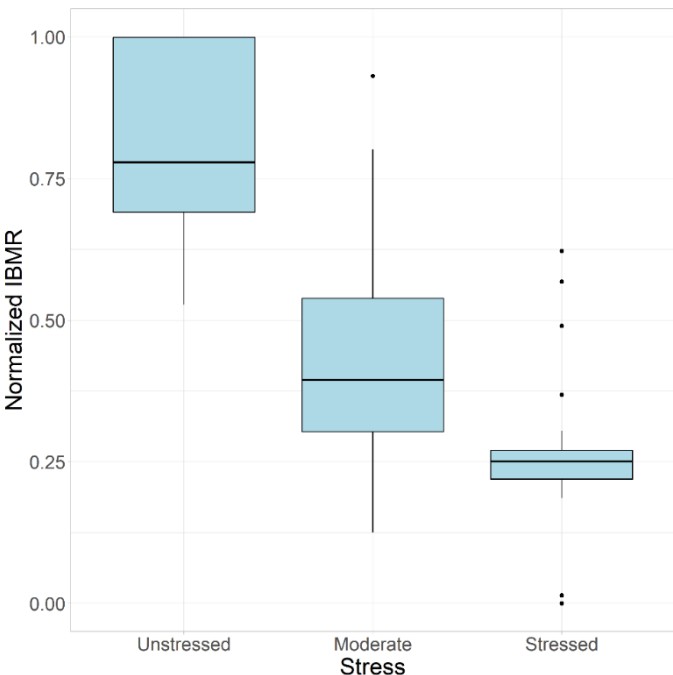

**Figure 3.** Boxplots of the normalized IBMR$_{GR}$ index among the unstressed, moderately stressed and stressed river reaches.

A simple Spearman correlation analysis between the IBMR$_{GR}$ index and the PC1 scores revealed a significant negative correlation (r = −0.81) which indicates a decline in the macrophyte index with a simultaneous increase of the hydromorphological stress. A linear regression showed a rather strong negative relationship (R$^2$ = 0.63) between the biological indicator (IBMR$_{GR}$) and the stressor (PC1 scores) (Figure 4).

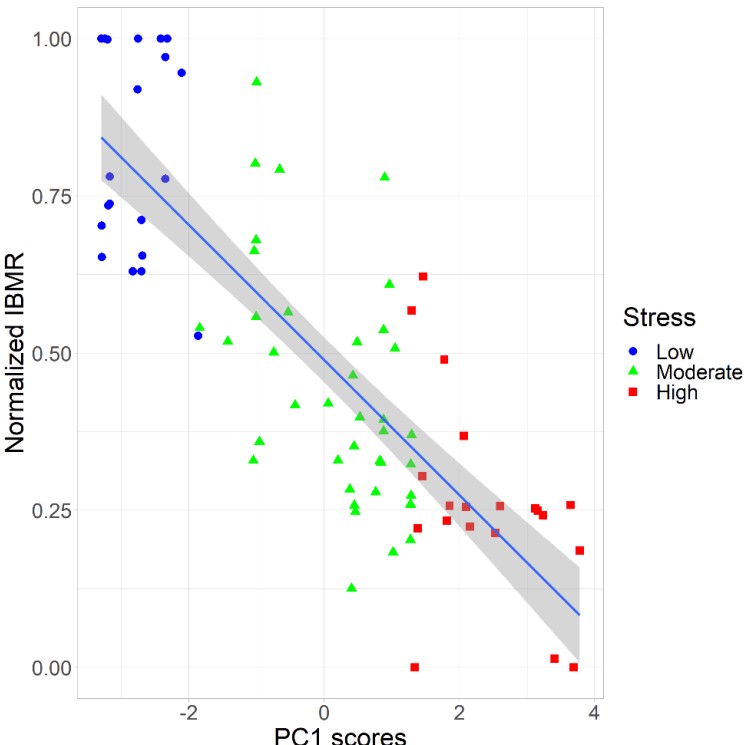

**Figure 4.** Linear regression between the normalized IBMR$_{GR}$ index and the scores of PC1. Higher PC1 values indicate higher levels of stress. Shaded area represents the 95% confidence intervals.

### 3.2. Definition of the Ecological Class Boundaries of the IBMR$_{GR}$

The next step was the definition of the ecological class boundaries of the IBMR$_{GR}$ index, so the index is WFD compliant. For this purpose the index was first normalized [30] to a range from 0 to 1. The boundary value of the normalized index between the High and the Good ecological quality class is the 25th percentile of the unstressed sites (0.69). The boundaries between the other quality classes are derived by dividing the 25th percentile of the unstressed sites (0.69) to the four remaining quality classes. Thus, the boundary between Good and Moderate is 0.518, between Moderate and Poor is 0.345 and between Poor and Bad is 0.173 (Figure 5).

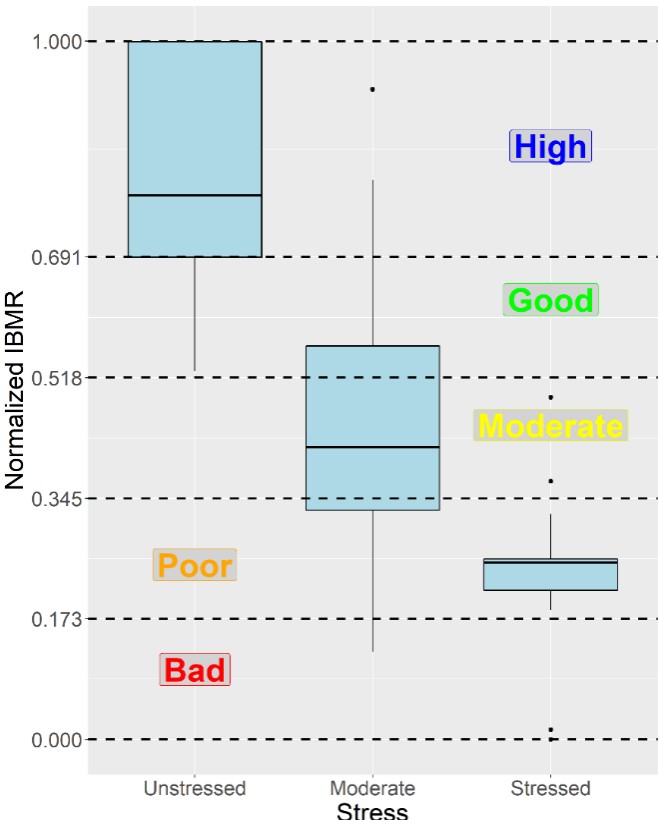

**Figure 5.** Ecological quality class boundaries of the normalized IBMR$_{GR}$ index.

### 3.3. Ecological Classification of the River Sites

The implementation of the index classified 15 sites as Good and 19 sites as High ecological quality classes. Thus, 45 from the total 79 sites failed the target of the Good quality class having either Bad, Poor or Moderate with the majority of them (27) classified as Poor (Figure 6). These results are generally in agreement with classification schemes based on other BQEs (e.g., benthic invertebrates and diatoms) [18,32] that have shown that a large share of rivers and streams in Greece is classified with less than Good ecological quality. This pattern is generally found in the whole of Europe where 40 to 50% of water bodies have failed the target of the Good ecological quality [33,34]. Currently there is a long discussion on why European freshwaters have not improved after the implementation of the measures proposed by the River Basin Management Plans and the Programmes of Measures [34,35], but deciphering the causes is a quite complex matter and beyond the scope of this study.

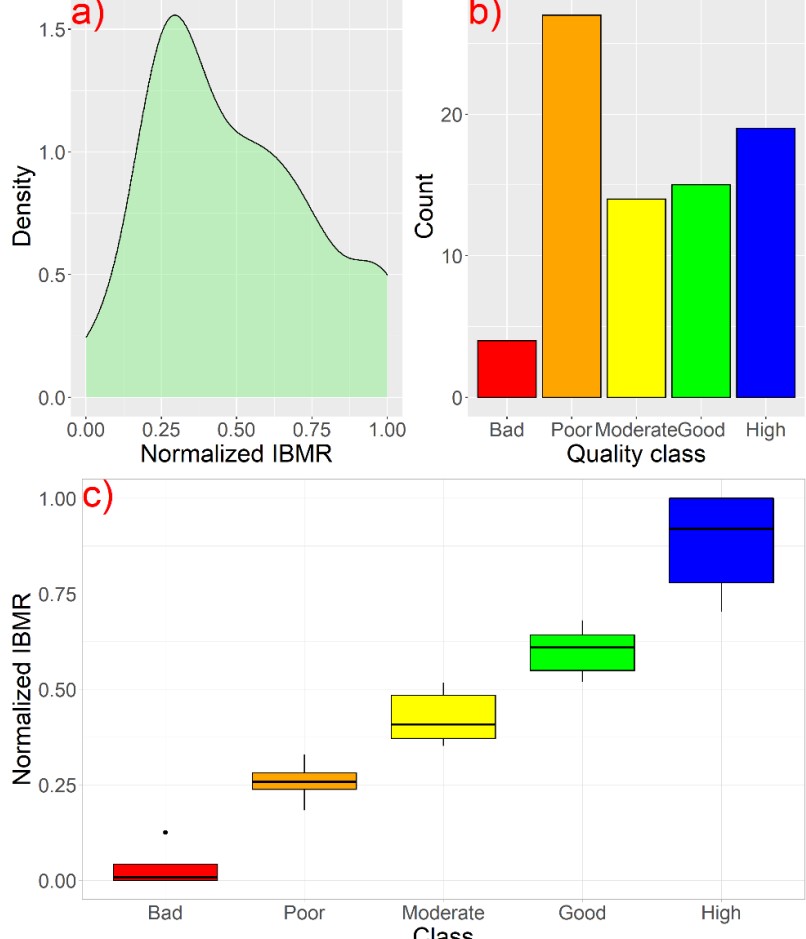

**Figure 6.** (**a**) Distribution of the normalized IBMR$_{GR}$ and (**b**) histogram of the ecological quality classes for all the studied reaches. Both plots show a high frequency of low values of normalized IBMR$_{GR}$ and Poor ecological quality. More than half of the river reaches have failed the Good ecological quality threshold. Subplot (**c**) shows boxplots of the normalized IBMR per class of ecological quality.

Since the WFD requires from the member states to differentiate the water bodies to types and establish type-specific reference conditions [36], ecological status assessment must be fulfilled for each type separately. Member states that share the same eco-region use a harmonized typology. For the Mediterranean region there are currently six intercalibration river types which are described as R-M1: Small, medium altitude Mediterranean streams with strong seasonal flow; R-M2: Small-medium lowland Mediterranean streams; R–M3: Large Mediterranean streams with strong seasonal flow; R-M4: Small–medium Mediterranean mountain streams with strong seasonal flow; R-M5: Small lowland temporary streams with temporary flow and VL: Very large rivers [32].

A first attempt to assess the ecological quality of the sites per intercalibration river type was made only for those types that were represented by a sufficient number of sites. The total of the 93 sites that were considered initially for the ecological assessment with the use of macrophytes are classified into three distinct river types, R-M1, R-M2 and R-M3. The majority (52) are characterized as R-M2, 35 sites as R-M1 and only 6 sites are classified as R-M3. Thus, the statistical analysis that involves the definition of a stressor gradient and quality class boundaries was only feasible for the types R-M1 and R-M2.

The methodology is the same as described in Sections 2.2 and 2.3. A MVAOS PCA was conducted separately for each type to identify the main stressor gradient and to distinguish the less disturbed or unstressed sites (Figure 7). Then the quality class boundaries are defined as described previously (Table 3).

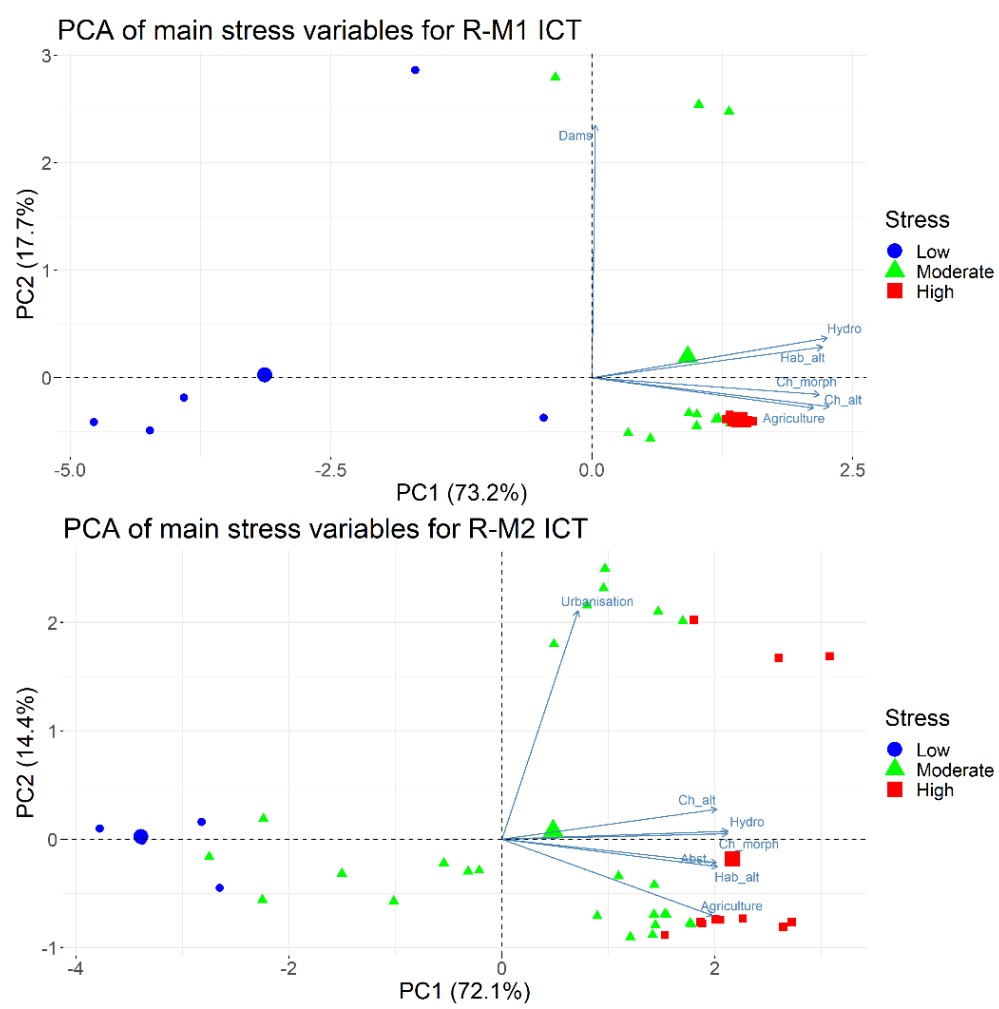

**Figure 7.** PCA biplots for sites belonging to ICT R-M1 (**top**) and R-M2 (**bottom**). PC1 components indicate a strong gradient of hydromorphological stressors. Unstressed sites (red dots) are positioned on the left part of the plot which, means low correlation with the stressor gradient.

**Table 3.** Quality class boundaries of the normalized IBMR defined for two MedGIG intercalibration river types (R-M1 and R-M2).

| Type | Ecological Quality Class Boundaries | | | | | | | | | |
|---|---|---|---|---|---|---|---|---|---|---|
| | **High** | | **Good** | | **Moderate** | | **Poor** | | **Bad** | |
| | *MIN* | *MAX* | *MIN* | *MAX* | *MIN* | *MAX* | *MIN* | *MAX* | *MIN* | *MAX* |
| **R-M1** | >0.705 | 1 | >0.529 | ≤0.705 | >0.352 | ≤0.529 | >0.176 | ≤0.352 | 0 | ≤0.176 |
| **R-M2** | >0.754 | 1 | >0.567 | ≤0.754 | >0.378 | ≤0.567 | >0.189 | ≤0.378 | 0 | ≤0.189 |

The linear regressions between the $IBMR_{GR}$ and the PC1 scores for each river type (R-M1 and R-M2) showed relatively good relationships with $R^2$ of 0.692 and 0.632 respectively (Figure 8). Most of the R-M1 sites (9 of 31) were classified as Bad, whereas only 6 sites met the target of the Good ecological quality class. Nine sites were not classified because only two or less macrophyte species with a specific score CS were recorded. For R-M2 sites, 22 of the 47 assessed sites had a Good or High quality class, 10 were classified as Moderate and 15 as Bad and Poor. In general, R-M2 sites showed better ecological quality conditions than the R-M1 sites.

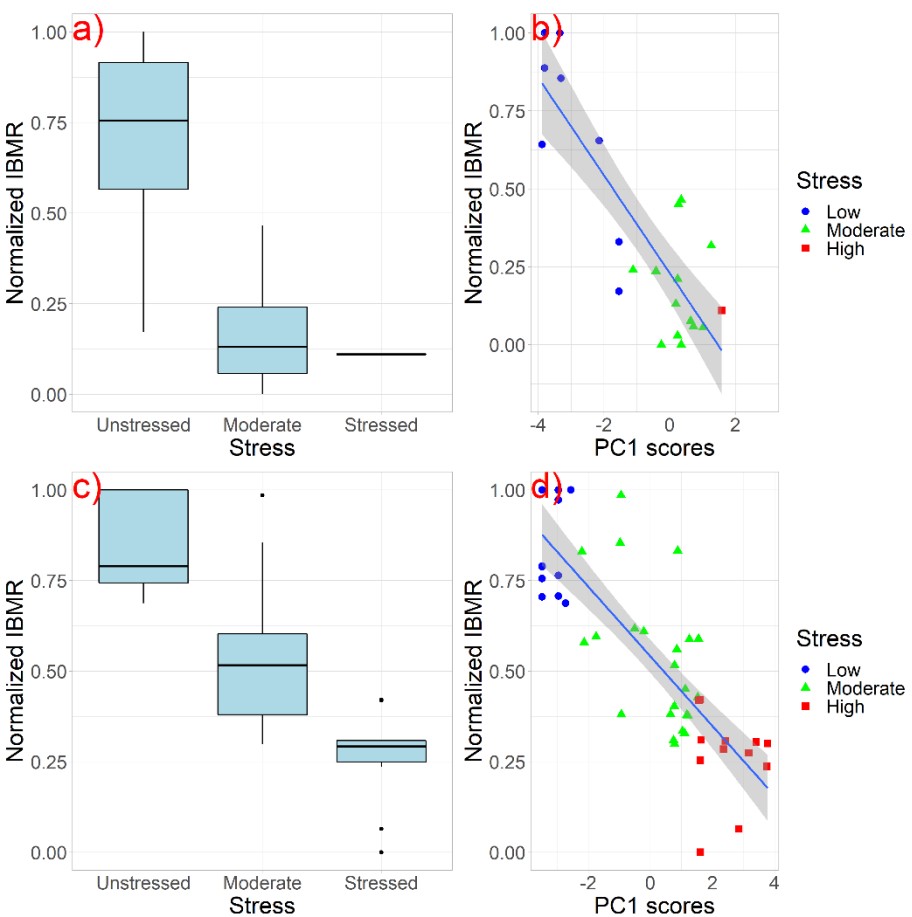

**Figure 8.** Boxplots of normalized IBMR$_{GR}$ index per stress level for R-M1 sites (**a**) and R-M2 sites (**c**). Linear regressions between normalized IBMR$_{GR}$ and PC1 scores of the main hydromorphological stress gradient as shown for R-M1 sites (**b**) and R-M2 sites (**d**).

### 3.4. The Use of the IBMR$_{GR}$ Index for Ecological Classification of Greek Running Waters—Strengths and Potential Caveats

The assessment of riverine ecological quality with the IBMR is a quite common method applied in several EU member states [13,37]. Although the index was primarily built to reflect the macrophyte responses to organic and trophic pollution gradients [17], in our case it was shown to correlate positively with a gradient of hydromorphological alterations. In Greece, hydromorphological modifications, such as bank and channel resectioning and realignment, are common stressors that occur in many river courses [38]. On many occasions, they are attributed to expansion of the agricultures [38] and other human activities and may co-occur with other perturbations, including point-sources of organic pollution [21]. This finding corroborates the results of previous studies done in different types of running water-bodies [39,40]. It is highly likely that hydromorphologically perturbed sites are also polluted. In our case, the PCA of the main stressor metrics showed that some of the most impaired sites were also related with the second PC component, which represents a gradient of ammonium, nitrate and phosphate concentrations in water. Still, we should bear in mind that the presented results derive from the pilot application of the index in rivers in Greece, and, as such the total number of tested sites is considered relatively small to exclude ultimate conclusions. An overall assessment of all the sites, regardless their typology, revealed rather promising results but further testing on discrete river types requires extensive collection of field data not only on macrophytes but mostly on environmental descriptors of key stressors. Here, we were able to test the IBMR$_{GR}$ index on two MedGIG types (R-M1 and R-M2) and we found relatively strong relationships between the index and the stressor gradient. These two types refer to mid-altitude or

lowland streams which are characterized by seasonal flows and show diverse aquatic plant communities [2]. For other river types, such as very large rivers and temporary streams, the application of the index might be problematic. For instance, macrophyte sampling at very large rivers or rivers of the type R-M3 might be less effective due to limitations associated with the large channel dimensions (depth and width) or muddy and clay substrate conditions. In addition, large rivers are more likely to be regulated and present frequent hydropeaking which although it plays a major role in shaping riparian plant communities [40–42], it may not be considered as a key stressor by the ecological assessment schemes. This is not a problem only for macrophytes but for other BQEs as well. Especially for countries where it is difficult to define sites with "reference conditions" different and additional approaches might be optimal [43,44]. Low land large rivers for example are more likely to be degraded and present a short gradient of pollution which makes it very challenging to identify sites with near natural conditions in terms of organic and nutrient pollution. For temporary streams, indicator species during low flow and dry conditions might be absent [4], which makes non-feasible the estimation of the index [45]. If there is not a sufficient number of indicator species the index will produce an unreliable result [1]. In our case, several sites were excluded from the ecological assessment because of a low number of indicator species. To deal with this issue, indicator species lists should be adjusted to include more new species that are characteristic to the local conditions [1,46]. Overall, the results of our research show that the macrophyte index IBMR$_{GR}$ can be used as a reliable indicator for the biological assessment of water quality and therefore it is recommended for future use in river management planning. Despite these promising results, there is still a need for expanding the stressor gradient, including additional monitoring sites. Future studies could then allow a comparative assessment between indices from different BQEs for the same water bodies [47] which would provide useful insights about the implementation of the ecological monitoring and assessment of the riverine systems, and promote the knowledge exchange towards to an enhanced management of rivers and riparian zones [48].

**4. Conclusions**

With this article we presented the first ever detailed overview of the implementation of a biological index based on macrophytes for the ecological assessment of the riverine ecosystems of Greece according to the guidelines of the WFD 2000/60. We showed that the IBMR$_{GR}$ index for Greek rivers relates strongly with the main stressor gradient and can distinguish the monitoring sites according to the stress level. We also found that the hydromorphological modifications were the main environmental stressors that correlated strongly with the IBMR$_{GR}$, whereas physicochemical stressors were of lesser importance. From a management perspective this means that mitigation and restoration measures should prioritize the recovery of the hydromorphologial functionality in order to improve the ecological quality and the overall ecosystem status. Furthermore, the results from the ecological classification showed that almost 40% of the sites had Good or High ecological quality, which is close to what other classification schemes have shown. Still, a type-specific classification such as the WFD dictates was only possible for two Mediterranean river types, mostly because of the small number of sites for the other types that did not allow for reliable statistical treatments. Nevertheless, the overall results are promising and indicate that the IBMR$_{GR}$ index is an efficient and applicable index for the rivers of Greece. However, future research is needed to explore the feasibility of application to additional river types and to strengthen the relationship between the index and the stressor gradient by collecting more, and possibly new metrics that could expand the range of the gradient.

**Supplementary Materials:** The following are available online at https://www.mdpi.com/article/10.3390/w14182771/s1. Table S1: A checklist with the plant species found at the studied sites.

**Author Contributions:** Conceptualization, E.P., K.S.; methodology, E.P., K.S.; data analysis, E.P., K.S., D.T.; visualization, K.S.; writing—original draft preparation, K.S., E.P.; writing—review and editing,

K.S., E.P., G.D., M.S., D.T.; data curation, E.P., G.D., M.S., D.T.; funding acquisition, E.P. All authors have read and agreed to the published version of the manuscript.

**Funding:** This research was funded by European and National grants from the Hellenic Centre for Marine Research under the "Monitoring of ecological quality of Greek rivers for the Implementation of Article 8 of WFD 2000/60/EE: samplings and analyses of aquatic macrophytes" research project. The publication of this article has been financed by the Research Committee of the University of Patras, funding number 826/30.08.2022).

**Institutional Review Board Statement:** Not applicable.

**Informed Consent Statement:** Not applicable.

**Data Availability Statement:** Derived data supporting the findings of this study are available from the corresponding author (E.P.) on request.

**Acknowledgments:** We are grateful to the staff of Patras Laboratory of Ecology for their invaluable help during the field work and collection of the samples. We would also like to express our gratitude to the Research Committee of the University of Patras for the support on managing the project and related activities.

**Conflicts of Interest:** The authors declare no conflict of interest.

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
