# Peer review of "Ecological Quality Assessment of Greek Lowland Rivers with Aquatic Macrophytes in Compliance with the EU Water Framework Directive"

_water, doi:10.3390/w14182771_

Round 1

Reviewer 1 Report

1-Which water quality parameters related to the macrophytes (e.i., MIR, IBMIR, D, N) were considered for this case study? 

2-The introduction section needs motivations and novelty.

3-Literature review needs the most updated references.

Reliability assessment of water quality index based on guidelines of national sanitation foundation in natural streams: integration of remote sensing and data-driven models

A novel multiple-kernel support vector regression algorithm for estimation of water quality parameters

4-Why did not authors consider turbidity, BOD, and COD ??

5-Why did authors use PCA? There are other feature extraction models such as forward selection (FS), Gamma test (GT), and polynomial chaotic expression (PCE).

Author Response

We appreciate the time and effort that the reviewer has dedicated to providing his/her valuable feedback on our manuscript. In this revised version, we have been able to incorporate changes that address the reviewer’s concerns. We also have checked carefully the manuscript for grammar and/or syntax errors and we have implemented significant changes concerning the writing as suggested. In what follows, we reply point-by-point to all the questions and comments made by the referee.

1-Which water quality parameters related to the macrophytes (i.e., MIR, IBMIR, D, N) were considered for this case study? 

We thank the reviewer for this comment. Water was sampled and transferred to the laboratory for the chemical quantification of orthophosphates, inorganic nitrogen species (nitrate, nitrite, and ammonium concentrations in water), total inorganic nitrogen and total phosphorus. Electrical conductivity, water temperature, dissolved oxygen, and pH were measured in site with a portable multi-meter probe. The water quality parameters that were considered in the statistical analysis are also listed in Table 1.

2-The introduction section needs motivations and novelty.

We thank the reviewer for this comment. We added a few lines in the last paragraph of the introduction and we modified accordingly the text in order to emphasize the fact that the current article is the first ever description of the national ecological assessment method of Greek rivers using aquatic macrophytes and the first ever presentation of the ecological classification of the rivers in compliance with the EU Water Framework Directive. Please check the additions at L47-50, L74-77, L89-92.

3-Literature review needs the most updated references.

Reliability assessment of water quality index based on guidelines of national sanitation foundation in natural streams: integration of remote sensing and data-driven models

A novel multiple-kernel support vector regression algorithm for estimation of water quality parameters

We added the two references as suggested. We have also update the reference list with four additional more recent articles. These are references [11,12,39,48]

4-Why did not authors consider turbidity, BOD, and COD ??

This is a good point made by the reviewer. BOD and COD are very useful indicators of organic pollution. In our case, organic pollution is a considerable stressor in rare occasions, usually at streams impacted by waste water emissions or agro-industrial activities.  Most streams are  characterized by the presence of hydromorphological modifications and nutrient diffuse pollution. In addition, the accurate quantification of these parameters, particularly BOD, requires the immediate transfer (within a few hours) of water samples to the laboratory which in our case, due to the remote location of most of the studied sites,  it further burdens the whole process of quantification of BOD and COD. However, we use several other water quality parameters such as dissolved oxygen, ammonium, nitrite and nitrate that can serve as proxy indicators of organic pollution and provide some information about the status of organic loads.

Concerning turbidity, we have measured electrical conductivity and total dissolved solids which in general correlate strongly with turbidity.

5-Why did authors use PCA? There are other feature extraction models such as forward selection (FS), Gamma test (GT), and polynomial chaotic expression (PCE).

This is a good question. Indeed, there are many other data analysis methods that have applications in environmental science. In our case, because our dataset includes several ordinal variables, we used a MVAOS PCA which is an extension of PCA for ordinal data. Furthermore, we used PCA to reduce the initial data dimensionality before we proceed our analysis with the final dataset.

Reviewer 2 Report

The manuscript has well written and informative. However, the reviewer has some major concerns.

-Please describe what kind of method has been used to determine the ammonium, Nitrate, total phosphorus, total nitrogen, orthophosphates

-Please provide the land use data for the study sites

- In the abstract section, mention some quantitative results, like which stressor was most responsible for ecological quality deterioration

- What kind of species did you find during the survey? Please add this information as a supplementary file

- What is your recommendation to improve the ecological quality of Greek lowland rivers? Please mention it in the conclusion part. 

Author Response

We appreciate the time and effort that the reviewer has dedicated to providing his/her valuable feedback on our manuscript. In this revised version, we have been able to incorporate changes that address the reviewer’s concerns. We also have checked carefully the manuscript for grammar and/or syntax errors and we have implemented significant changes concerning the writing as suggested. In what follows, we reply point-by-point to all the questions and comments made by the referee.

The manuscript has well written and informative. However, the reviewer has some major concerns.

-Please describe what kind of method has been used to determine the ammonium, Nitrate, total phosphorus, total nitrogen, orthophosphates

We thank the reviewer for this comment. We added a mention in the methods section about the chemical analytical procedures that follow the standard APHA methodology.

-Please provide the land use data for the study sites

We thank the reviewer for this remark. We have to clarify that we do not have quantitative data of the various land use types (e.g., share percentages) for each site. Instead, we estimated qualitatively the dominant land use type in the wider adjacent area to the site, for example agricultural, natural or urban land uses. Thus, we preferred to not include and present this kind of information in the article

- In the abstract section, mention some quantitative results, like which stressor was most responsible for ecological quality deterioration

This is a very good point made by the reviewer. We added the following lines in the abstract:

“More specifically, the first principal component explained the 51 % of the total variance of the data, representing a moderately strong gradient of hydromorphological stress, whereas the second component explained the 22.5 %, representing a weaker gradient of physicochemical stress”

- What kind of species did you find during the survey? Please add this information as a supplementary file

We agree that it could be possible to include a species list as a supplementary material. Although we think that a list of all species found at the sites are beyond the scope of the article we added a table as a supplementary material.

- What is your recommendation to improve the ecological quality of Greek lowland rivers? Please mention it in the conclusion part.

We agree with the reviewer that a comment about the improvement of the ecological quality could be useful and interesting. Although we feel that recommendations about measures that may improve the overall ecological quality are beyond the scope of this article, we added a brief mention about management actions that could be undertaken at catchment scale and should have a positive effect on the ecology of the studied rivers.

Round 2

Reviewer 1 Report

Accept as is

Reviewer 2 Report

The authors have responded to my concern and improved the manuscript. The current form of the manuscript can be accepted for publication.